# Cholesterol Metabolites 25-Hydroxycholesterol and 25-Hydroxycholesterol 3-Sulfate Are Potent Paired Regulators: From Discovery to Clinical Usage

**DOI:** 10.3390/metabo11010009

**Published:** 2020-12-25

**Authors:** Yaping Wang, Xiaobo Li, Shunlin Ren

**Affiliations:** 1Department of Internal Medicine, McGuire Veterans Affairs Medical Center, Virginia Commonwealth University, Richmond, VA 23249, USA; Yaping.wang@va.gov; 2Department of Physiology and Pathophysiology, School of Basic Medical Sciences, Fudan University, Shanghai 200032, China; xbli@fudan.edu.cn

**Keywords:** oxysterol sulfates, oxysterol sulfation, epigenetic regulators, 25-hydroxysterol, 25-hydroxycholesterol 3-sulfate, 25-hydroxycholesterol 3,25-disulfate

## Abstract

Oxysterols have long been believed to be ligands of nuclear receptors such as liver × receptor (LXR), and they play an important role in lipid homeostasis and in the immune system, where they are involved in both transcriptional and posttranscriptional mechanisms. However, they are increasingly associated with a wide variety of other, sometimes surprising, cell functions. Oxysterols have also been implicated in several diseases such as metabolic syndrome. Oxysterols can be sulfated, and the sulfated oxysterols act in different directions: they decrease lipid biosynthesis, suppress inflammatory responses, and promote cell survival. Our recent reports have shown that oxysterol and oxysterol sulfates are paired epigenetic regulators, agonists, and antagonists of DNA methyltransferases, indicating that their function of global regulation is through epigenetic modification. In this review, we explore our latest research of 25-hydroxycholesterol and 25-hydroxycholesterol 3-sulfate in a novel regulatory mechanism and evaluate the current evidence for these roles.

## 1. Introduction

Oxysterols are the oxidized form of cholesterol. There are many review articles describing the origins and metabolism of oxysterols [1]. In vivo, enzymatic transformation of sterols to oxysterols is for the biosynthesis of important biological products such as steroid hormones, bile acids, and vitamin D in cells, blood, and tissues [2,3,4]. Most of the oxysterols are their metabolic intermediates, which do not have significant biological activity [5,6]. The potent active oxysterols include 25-hydroxycholesterol (25HC), which is synthesized by cholesterol 25-hydroxylase (CH25L) in hepatocytes and macrophages [7,8]; 27-hydroxycholesterol (27HC), synthesized by cholesterol 27-hydroxylase (CYP27A1: cytochrome P450 family 27 subfamily A member 1), in many cells [9]; 24-hydroxycholesterol (24HC), synthesized by cholesterol 24-hydroxidase (CH24L) (occurs mainly in brain) [10]; and 24, 25-epoxycholesterol, synthesized from cholesterol precursors by multiple enzymes [11]. These oxysterols participate in many biological processes including cholesterol homeostasis, triglyceride metabolism, inflammatory responses, cell proliferation, platelet aggregation, and apoptosis [12,13,14]. Oxysterols have also been implicated in many diseases such as metabolic syndrome and neurodegenerative diseases [15,16]. Oxysterols can be sulfated by sulfotransferase 2B1b (SULT2B1b) at the 3 position of ring A of cholesterol to be oxysterol 3-sulfates, including 25HC3S, 24HC3S, 27HC3S, as well as Xol3S (cholesterol 3-sulfate) [17,18,19], as shown in Figure 1. Oxysterol sulfate can be further sulfated by sulfotransferase 2A1 (SULT2A1) to be oxysterol disulfates [20,21,22]. The most eminent oxysterol sulfate is 25-hydroxycholesterol 3-sulfate (25HC3S), which can be further sulfated by SULT2A1 to 5-cholesten-3β, 25-diol-disulfate (25HCDS). 25HC3S and 25HCDS are the only oxysterol sulfates that have been identified in vivo in hepatocyte nuclei, while 27HC3S has been identified in human sera and 24HC3S has been identified in urine [19,23,24]. Interestingly, 25HC3S/25HCDS are also potent regulators but function in different directions from their precursor 25HC [22,23]. Recent reports have shown that 25HC and 25HC3S/25HCDS coordinately regulate important cell events, including maintenance of lipid homeostasis and responses to stress signals via epigenetic modification (data submitted for publication). Their roles in disease development and recovery, and their detailed molecular mechanism are the focus of the present review.

Cholesterol can be hydroxylated by CYP27A1 to 25HC or 27HC in the mitochondria and hydroxylated to 25HC by CYP3A4 (cytochrome P450 3A4) or by cholesterol 25-hydroxylase (CH25H) in the endoplasmic reticulum [25]. Cholesterol can also be hydroxylated by cholesterol 24-hydroxylase to 24HC in brain tissue. Cholesterol precursors can be used for synthesis of desmosterol via a shunt of the mevalonate pathway. Desmosterol can be oxygenated by CYP46A1 (cytochrome P450 46A1) to form 24, 25-epoxycholesterol (24,25EC) [26]. 25HC, 27HC, 24HC, as well as cholesterol can be subsequently sulfated at the 3β position by SULT2B1b to form 25HC3S, 27HC3S, 24HC3S, and Xol3S, respectively. Most likely, 24, 25EC can be sulfated to be 24, 25EC3S.

## 2. 25-Hydroxycholesterol Induces Lipogenesis and Cell Apoptosis

Cholesterol 25-hydroxylase (CH25L) in cytosol catalyzes the formation of 25-hydroxycholesterol (25HC) from cholesterol, leading to the repressed expression of cholesterol biosynthetic enzymes, and plays a key role in cell positioning and movement in lymphoid tissues [8,27]. 25HC is an intermediate in the biosynthesis of 7-alpha, 25-dihydroxycholesterol (7-alpha, 25-DC), an oxysterol that acts as a ligand for the G protein-coupled receptor GPR183/EBI2, a chemotactic receptor for a number of lymphoid cells [27]. CH25L may play an important role in regulating lipid metabolism by synthesizing a corepressor that blocks sterol regulatory element binding protein (SREBP) processing [8]. In testes, the production of 25-hydroxycholesterol by macrophages may play a role in Leydig cell differentiation [28]. 

25HC has been believed to be one of the most potent ligands of liver X receptor (LXRs) and plays an important role in the control of lipid metabolism [29]. LXR ligands upregulate the expression of cholesterol reverse transporters such as ABCA1 and ABCG1 via activation of LXR/RXR heterodimers. ABCA1/G1 is known to mediate efflux of cellular cholesterol and phospholipids, which is the target for therapy of anti-atherosclerosis [30]. Unfortunately, administration of synthetic ligands to mice triggers induction of the lipogenic pathway and elevates plasma triglyceride levels [31]. The addition of 25HC to human hepatocytes increases gene expression of key enzymes, acetyl CoA acetylase (ACC), and fatty acid synthetase (FAS) in lipid biosynthesis and increases intracellular lipids levels via the SREBP-1c pathway [32]. 25HC has also been described as functioning in the immune system, suppressing immunoglobulin production in B cells [9], and inducing the expression of inflammatory cytokine interleukin-8 [33,34,35,36]. 25HC has been shown to directly restrict target cell entry of enveloped viruses by inhibiting fusion of virus and cell membranes [37,38,39,40,41]. However, LXR targeting molecules are very limited and cannot be used to explain the function of 25HC global regulation in lipid metabolism, inflammatory responses, and cell apoptosis, indicating that another mechanism should exist.

25HC plays an important role in lipid accumulation in hepatocytes and development of nonalcoholic fatty liver diseases (NAFLD) [42,43]. 25HC regulates excess acetyl CoA for synthesis of lipids as energy storage. For example, carbohydrates such as fructose and glucose can be hydroxylated to acetyl-CoA, the major substrate for producing energy, ATP, in vivo. When ATP concentrations reach high levels, the ratio of AMP/ATP decreases and excess acetyl-CoA is shunted into synthesis of cholesterol and oxysterol such as 25HC. 25HC induces biosynthesis of free fatty acids and triglycerides. Lipid accumulation in hepatocytes induced by high glucose media has been widely used as in vitro model for the study of NAFLD [43,44]. A recent report has shown that, when human hepatocytes were incubated in media containing high glucose levels, 25HC was significantly increased in their nuclei [43]. Meanwhile, the high glucose incubation increases CpG (5′—cytosine—phosphate—guanine—3′) methylation and increases in ^5m^CpG levels in promoter regions and silences the expression of many key genes involved in PI3K, cAMP, insulin, insulin secretion, diabetic, and NAFLD signaling pathways, leading to lipid accumulation [43]. Interestingly, the sulfated 25HC, 25HC3S converted the methylated CpG in the regions, which has been shown as a distinct yet potent regulator of cellular functions [19].

## 3. Discovery of Oxysterol Sulfates and Exploration of Their Function

Bile acids are synthesized from cholesterol via two pathways: the classic pathway and the alternative pathway [45,46,47]. The primary pathway for biosynthesis of bile acids in humans is the classic pathway. Cholesterol 7α-hydroxylase (CYP7A1), located in the microsome, is the rate-limiting step of the classic pathway [47,48,49,50]. The secondary pathway for biosynthesis of bile acids in humans is the alternative pathway. Cholesterol delivery protein StarD1, which delivers cholesterol to the mitochondria, is the rate-limiting step in the alternative pathway [51]. Regulatory oxysterol intermediates, such as 25- or 27-hydroxycholesterol (25HC and 27HC), have been shown to genetically regulate cholesterol homeostasis, including many genes that encode proteins and enzymes involved in cholesterol metabolism and transportation [52,53,54,55]. By creating overexpressed StarD1 in vivo models, it was discovered that the rate of bile acid synthesis matched the rate of bile acid synthesis in overexpressed CYP7A1 in vivo models [56]. The results indicate that StarD1, which delivers cholesterol to the mitochondria, is the rate-limiting step in the alternative pathway. By manipulating StarD1, exploration of the function of the alternative pathway was made possible. [^14^C]-Cholesterol derivatives were extracted and purified from the mitochondria, cytosol, and nucleus following StarD1 overexpression. Certain water-soluble [^14^C]-oxysterol products from the nucleus were isolated and purified by chemical extraction and HPLC (high-performance liquid chromatography). Enzymatic digestion, HPLC, and MS/MS (mass/mass spectrometry) analysis identified the water-soluble oxysterol as 5-cholesten-3β, 25-diol 3-sulfate (25HC3S) and 5-cholesten-3β,25-diol 3β,25-disulfate (25HCDS) [24,57]. Although many oxysterols can be sulfated to oxysterol sulfates, 24-hydroxycholesterol 3-sulfate (24HC3S), 27-hydroxycholesterol 3-sulfate (27HC3S), and cholesterol 3-sulfate (Xol3S), 25HC3S/25HCDS are the only sulfated oxysterols discovered in hepatocyte nuclei and seems to be the most potent regulator [23]. 

Steroid sulfatase (STS) is the enzyme most likely responsible for degradation of oxysterol sulfates. STS, formerly known as arylsulfatase C, is a sulfatase enzyme involved in the metabolism of steroids. The protein encoded by the *STS* gene catalyzes the conversion of sulfated steroid precursors to the free steroid. It was noted that high levels of oxysterol sulfates have been found in the serum of patients with STS deficiency, indicating that the oxysterol sulfates may be degraded by the sulfatase [58]. Further studies are needed to confirm whether STS is responsible for the degradation of 25HC3S/25HCDS and participate in regulation of their function. 

## 4. 25HC3S/25HCDS Suppresses Cholesterol and Triglyceride Biosynthesis in Hepatocytes

LXRs are members of the nuclear receptor superfamily of ligand-activated transcriptional factors involved in the regulation of lipid metabolism and inflammatory process [59]. LXRα is a key transcriptional regulator of lipid metabolism in hepatocytes. LXRα increases the expression of SREBP-1c, which in turn regulates at least 32 genes involved in lipid synthesis and transport [57,60,61]. The addition of 25HC3S or 25HCDS markedly decreased nuclear LXRα protein levels, followed by decreases in SREBP-1c mature protein and mRNA levels in hepatocytes and macrophages [26,33]. 25HC3S/25HCDS also decreased the expression of SREBP-1c/SREBP-2 responsive genes, including ACC-1, FAS, GPAM, HMGR, LDLR, etc. As a result, 25HC3S and 25HCDS decreased fatty acid, triglyceride, and cholesterol biosynthesis. In contrast to 25HC3S/25HCDS, 25HC acted as an LXR agonist and increased SREBP-1c and FAS mRNA levels [18,32,60,62]. 

## 5. 25HC3S Increases Nuclear PPARγ Levels and Suppresses Inflammatory Responses via the PPARγ/IκBα Signaling Pathway 

The excess amounts of free fatty acids and oxysterols inside the hepatocytes are thought to be the culprits for causing lipotoxicity, activating the inflammatory signaling pathways and ultimately leading to cell death [61,63,64]. PPARγ is particularly important in regulating lipid metabolism and inflammatory responses. Activation of PPARγ uregulates IκBα levels, decreases NF-κB nuclear translocation, and consequently represses NF-κB-dependent genes such as IL1β and TNFα [65]. The addition of 25HC3S and rosiglitazone (synthetic PPARγ agonist) significantly increased nuclear PPARγ levels, while T0070907 (synthetic antagonist) and 25HC decreased the levels in human hepatocytes and THP-1 macrophages. These results suggest that 25HC and 25HC3S regulate inflammatory responses via the PPARγ/IκBα signaling pathway but in the opposite direction [18,32,61,66]. 

## 6. Biochemical Mechanism: 25HC and 25HC3S Are Paired Epigenetic Regulators

The function of 25HC and 25HC3S in global regulation indicates that they are epigenetic regulators. Methylation at position 5 of cytosine (5-methylcytosine, ^5m^C) in DNA promoter regions is an important epigenetic modification that regulates gene expression and other functions of the genome [43]. Cytosine methylation of CpG in promoter regions is inversely correlated with the transcriptional activity of associated genes as it causes chromatin condensation and thus gene silencing [67]. Previously published literature has demonstrated that dysregulation of CpG methylation and gene expression is important in metabolism and can affect tissue function and in turn the metabolic state [68]. It has been shown that cytosine methylation is catalyzed by DNA methyltransferases (DNMT-1, 3a/3b), which have been reported to play an important role in the regulation of DNA methylation/demethylation [69,70]. Recent reports uncovered that 25HC and 25HC3S are ligands of DNA methyltranferase-1 (DNMT-1). 25HC specifically activates DNMT-1 activity up to 8-fold, and 25HC3S inactivates DNMT-1, 3a, and 3b down to less than 5% at µM concentrations [43]. Thus, 25HC and 25HC3S, novel endogenous and cellular regulatory molecules, epigenetically regulate lipid metabolism, cell survivals/death, and inflammatory responses via DNA CpG methylation and ^5m^CpG demethylation. High glucose incubation increases CpG methylation in promoter regions via increasing nuclear 25HC levels, which silences key gene expressions involved in PI3K-Akt, cAMP, NAFLD, type II diabetes mellitus, and insulin secretion signaling pathways [43]. The results strongly indicate that the epigenetic modification is responsible for the development of metabolic syndrome, including atherosclerosis, obesity, NAFLD, and type II diabetes. Interestingly, the sulfated oxysterol 25HC3S demethylates ^5m^CpG in these promoter regions, increases these gene expressions, and upregulates these master signaling pathways. 25HC3S regulates the signaling pathways in an opposite direction from its precursor 25HC. The results from these studies indicate an important regulatory mechanism by which intracellular oxysterols and oxysterol sulfates cooperatively regulate critical cell signaling pathway response to stress responses. Oxysterol sulfation appears to play important physiological and pathophysiological roles as protein phosphorylation, inositol phosphorylation, and sphingosine phosphorylation in regulating cellular functions [63]. Different from phosphorylation, oxysterol sulfation globally regulates gene expression at the transcriptional levels. The proposed mechanism is shown in Figure 2. The global potent regulatory functions of 25HC3S, such as decreases in lipid accumulation, suppresses of inflammatory responses, and promotion of cell survival [63], suggest its clinical utility in the treatment of metabolic syndromes or diseases.

There are two pathways, classic and alternative pathways, for cholesterol degradation and bile acid synthesis in hepatocytes. The classic pathway is dominant for bile acid biosynthesis, and the alternative serves as a regulatory pathway involved in lipid metabolism and cell proliferation. When cholesterol levels increase by exogenous diet or endogenous biosynthesis in the cells, (1) StarD1 delivers cholesterol into mitochondria where CYP27A1 is located; (2) cholesterol is 25-hydroxylated by CYP27A1 to form 25-hydroxycholesterol and sulfated at its 3β position to form 25-OH cholesterol 3β-sulfate (25HC3S) by hydroxycholesterol sulfotransferase 2B1b (SULT2B1b), and 25HC3S can be further sulfated to 25HCDS by SULT2A1; and (3) oxysterols and sulfated oxysterols might play important roles in maintenance of lipid homeostasis, inflammatory responses, apoptosis, and cell proliferation by epigenetic regulation. Oxysterol sulfates such as 25HC3S decrease lipid accumulation, anti-inflammatory responses, and antiapoptosis by increasing gene expression through demethylation of ^5m^CpG in promoter regions of the key genes involved in MAPK-ERK and calcium-AMPK signaling pathways, such as CREB5 (CAMP Responsive Element Binding Protein 5), BAD (BCL2 Associated Agonist of Cell Death), and ERK (Mitogen-activated protein kinase 1).

## 7. In Vivo Studies: 25HC3S/25HCDS Has Potential for Clinical Use

25HC3S as an endogenous and orally bioavailable epigenetic regulator has been shown in nonclinical studies to play an important regulatory role in lipid homeostasis, inflammation, and cell survival, which have strong clinical significance, as shown Figure 3.

The products of oxysterol sulfation, 25HC3S (DUR928) and 25HCDS, suppress lipogenesis and apoptosis in hepatocytes, inhibit expression of inflammatory factors in macrophages, and improve hepatocyte proliferation. These beneficial effects help alleviate NAFLD and acute organ injury and promote liver regeneration.

## 8. 25HC3S Decreases Lipid Accumulation, Suppresses Inflammatory Responses, and Decreases Liver Fibrosis in Mouse NAFLD Models 

The liver is a key metabolic organ that plays a central role in lipid homeostasis [71]. Prolonged consumption of high-fat diets elevates plasma FFA (free fatty acids) levels, which induces intrahepatocellular accumulation of triglycerides and diacylglycerol. Eventual ballooning of lipid-laden hepatocytes will produce low-degrade inflammation through the activation of NFκB. NFκB incites a cascade of pro-inflammatory reactions that will produce inflammatory cytokines, such as IL-6 [72]. The accumulation of lipid droplets in hepatocytes becomes the hallmark of NAFLD. NAFLD is a spectrum of liver pathology that ranges from steatosis, steatohepatitis (NASH), to cirrhosis. The pathogenesis of NAFLD is a long and complex process. It involves many epigenetic and genetic factors [73]. By exploring and manipulating the genetic and epigenetic factors, uncharted cholesterol pathways as well as potential innovative treatments for these liver pathologies may be discovered. 

Overexpression of SULT2B1b in mice fed with a high fat diet (HFD) significantly increased oxysterol sulfate formation in the liver [74]. Higher oxysterol sulfate concentration resulted in inhibition of the LXRα/SREBP-1c signaling pathway, which lowered circulating and hepatic lipid levels. Interestingly, this phenomenon only occurs with adequate levels of 25HC. Without 25HC, overexpression of SULT2B1b in mice fed with HFD failed to increase oxysterol sulfate formation in the liver, which failed to lower circulating and hepatic lipid levels. Similar effects were also observed in LDLR knockout mice. This indicates that the net mechanism of SULT2B1b is the inhibition of lipid synthesis rather than clearance of lipid stores. 

Administration of 25HC3S reduced lipid accumulation, improved insulin resistance, and suppressed inflammation in vitro and in vivo [18,32,57,60,61,62,63,64]. Several studies have demonstrated that pro-inflammatory cytokines, such as TNFα, IL-1, and IL-6, are the major contributors to the progression from steatosis to NASH [75,76]. Pro-inflammatory cytokine concentrations are directly correlated with the NAFLD activity score in NASH and the degree of fibrosis [77,78]. 25HC3S significantly suppresses TNFα-induced inflammatory response in HepG2 cells and in LPS-induced inflammatory response in THP-1 microphage [64,66]. 

A NASH mouse model was established by subcutaneous injection of 200 µg streptozotocin solution 2 days after birth in male C57BL/6J mice, followed by high-fat diet (57 kcal % fat) feeding at 4 weeks of age until the end of the study [79]. 25HC3S was tested in 2 stages. During the first stage, NASH mice were treated with daily oral doses of 10 or 50 mg/kg of 25HC3S for four weeks. There was a significant decrease in NASH development and significant reduction in TNFα concentration in hepatocytes. During the second stage, NASH mice were treated after liver fibrosis was established. Daily oral administration of 25HC3S for four weeks significantly decreased liver fibrosis, hepatocyte ballooning, and expression of collagen Iα1 and IIIα1.

25HC3S administration in HFD mice also decreases ectopic lipid droplet accumulation in hepatocytes and ultimately improves insulin signaling and reduces insulin resistance [60]. SULT2B1b transgenic mice showed significantly better insulin tolerance and glucose tolerance compared to HFD-fed wild-type mice. More interestingly, the liver tissues in SULT2B1b transgenic mice are resistant to lipid accumulation. Hepatic triglyceride levels are significantly lower (approximately 50%) than triglyceride levels in wild-type mice [80]. On the flipside, SULT2B1b null (SULT2B1b^−/−^) mice showed increased gluconeogenic gene expression and elevated fasting glucose level [81]. 

## 9. 25HC3S and Oxysterol Sulfation Suppress Hepatocyte Apoptosis and Promotes Regeneration in Mouse Hepatectomy Model

Hepatocytes are unique parenchymal cells because they retain a stem cell-like ability to regenerate [73]. This property underlines the remarkable capacity of the liver to regenerate following acute injuries that diminish hepatic mass. Hepatocyte regeneration is also an important adaptive response in chronic liver diseases as hepatocytes destroyed by lipid accumulation, inflammation, and other injuries are replaced to maintain adequate liver function. The factors that regulate hepatocyte regeneration have not been fully identified but are thought to include growth factors and cytokines arising from both extrahepatic sites and the liver. The ability of quiescent adult hepatocytes to replicate in response to physiologic stimuli is a key point to promote proliferation of these cells [73,82]. 25HC3S promotes hepatocyte regeneration and the prevention of cell death by suppressing apoptotic gene expression [83,84]. Profiler™ PCR array and RT-qPCR analysis showed that both exogenous administration of 25HC3S and endogenous biosynthesis of 25HC3S (generated by overexpression of SULT2B1b plus administration of 25HC) significantly upregulated the expression of Wt1, PCNA, cMyc, cyclin A, FoxM1b, and CDC25b in a dose-dependent manner. At the same time, 25HC3S substantially downregulated the expression of cell cycle arrest gene Chek2 and apoptotic gene Apaf1. Furthermore, 25HC3S significantly induced hepatic DNA replication, measured by immunostaining of the PCNA labeling index. Double immunofluorescence confirmed that PCNA expression in the nuclei was accompanied by SULT2B1b in the cytosol; no PCNA expression was detected in the cells without SULT2B1b expression, indicating that SULT2B1b is responsible for the expression of PCNA [84]. These results confirm that 25HC3S promotes cell regeneration, indicating potential benefits in aiding the recovery of injured organs, such as those in acute organ failure.

## 10. 25HC3S/25HCDS Alleviate LPS—Induced Acute Organ Injury in Mouse Model 

25HC3S and 25HCDS have been shown to be able to suppress inflammatory responses, inhibit apoptosis, and promote hepatocyte proliferation [64,66,83]. Administration of 25HC3S/25HCDS alleviates injured liver, lung, and kidney function and decreases mortality in LPS-induced mouse models [66]. Reduction of mortality by 25HC3S/25HCDS in animal models is striking. When administered 25HC3S within 96 h to animals with LPS-induced organ injury, the survival rate was 90% compared to the survival rate of 10% in animals with LPS-induced organ injury that was not administered 25HC3S [66]. 

Time-course studies of apoptotic gene expression following LPS challenge in 25HC3S-treated mice were performed to identify whether the 25HC3S-induced anti-inflammatory effects are preventive or therapeutic [66]. The mRNA levels of genes involved in apoptosis were determined using the RT^2^ Profiler PCR Array. Cluster gram analysis showed that the profiles were very similar between control and treated groups at 3 and 6 h. Interestingly, however, a significant difference was observed at 20 h. The profiles of the treated group at 20 h are more like to the normal profile. Scatter plot analysis showed that expression of only one gene was significantly decreased and one significantly increased between the treated and untreated groups at 3 h, that one gene decreased and two genes increased at 6 h, and that 19 genes increased and 6 genes decreased at 20 h. The genes that were affected the most were involved in immune system processes, autophagy processes, and apoptosis. The correlation was further confirmed by the levels of serum cytokines (IL-6, TNFα, and IL-1β). These results provide evidence that 25HC3S is therapeutic and not preventative. 

## 11. Clinical Development: 25HC3S to Serve as a Treatment in Chronic Liver Diseases 

Clinical-enabling toxicology and pharmacokinetics (PK) of DUR-928 (25HC3S) were evaluated in both acute and repeat-dose studies, up to 28 days in duration, in multiple species, including mice, rats, rabbits, dogs, and primates [85,86]. In these studies, DUR-928 was administered at the highest feasible doses via oral gavage or capsule, IM or SC injections, or IV infusion. There were no drug-related adverse effects observed for all doses in these studies to date, supporting the use of DUR-928 in humans.

## 12. Phase 1 Study: Safety and Pharmacokinetics of DUR-928 in Healthy Human Volunteers

Oral or parenteral administration of DUR-928 has been evaluated for safety and pharmacokinetics in 97 healthy human volunteers to date. DUR-928 was given either as a single dose or as a daily repeat dose for up to 5 days. DUR-928 was readily absorbed orally, showing a dose-dependent absorption. When DUR-928 was given parenterally (either IM injection or IV infusion), the systemic drug levels were proportional to the dose. The plasma half-life of DUR-928 was 1–3 h for both oral and parenteral routes, and there was no drug accumulation after daily repeated dosing. DUR-928 was well tolerated among healthy volunteers at all dose levels evaluated in all these studies [86,87]. 

## 13. Phase 1b Study: DUR-928 in Liver Function-Impaired (NASH) Patients

Phase 1b clinical trials were open-label and single-ascending-dose studies to assess safety and pharmacokinetics in liver function impaired (cirrhotic and non-cirrhotic NASH) patients and in normal liver function-matched control subjects (MCS). MCS were matched by age, body mass index, and gender. There were two dose groups (a low dose of 50 mg and a high dose of 200 mg administered orally), with each dose group having 10 NASH patients and 6 MCS. All NASH patients and MCS tolerated DUR-928 well. In both the low- and high-dose groups, the pharmacokinetic parameters were comparable between NASH patients and MCS. Following low and high doses of oral DUR-928, the systemic drug levels were dose dependent [79].

While this study was not designed to assess efficacy, statistically significant reductions from baseline of certain biomarkers after a single oral dose of DUR-928 was observed, which included both full-length (M65) and cleaved (M30) cytokeratin-18 (CK-18), bilirubin, high sensitivity C-reactive protein (hsCRP), and IL-18. The mean reduction of full-length (M65) CK-18 (a generalized cell death marker) at the measured time point of the greatest effect (12 h after dosing) was 33% in the low dose cohort and 41% in the high dose cohort, while the mean reduction of cleaved (M30) CK-18 (a cell apoptosis marker) at the measured time point of the greatest effect (12 h after dosing) was 37% in the low-dose cohort and 47% in the high-dose cohort. The mean reduction in total bilirubin at the measured time point of the greatest effect (12 h after dosing) was 27% in the low-dose cohort and 31% in the high-dose cohort. The greatest reductions of hsCRP, a marker of inflammation, and IL-18, an inflammatory mediator, were observed at 24 h and 8 h after dosing, respectively [79]. The results indicate that DUR928 (25HC3S) has potential to treat the notorious diseases, NAFLD.

## 14. Perspective

Chronic systemic metabolic disorders include obesity, hyperlipidemia, atherosclerosis, diabetes, NAFLD, and NASH. Millions of people in the world suffer from these diseases, which are the number one causes of human deaths. The majority of therapies developed for the treatment of metabolic syndrome are agonists or antagonists for specific enzymes, receptors, or transporters. Statins inhibit 3-hydroxy-3-methylglutaryl coenzyme-A reductase (HMGR) to reduce blood cholesterol levels. An autoantibody against proprotein convertase subtilisin/kexin type 9 (PCSK9) prevents binding to and thus prevents inhibition of endothelial LDL receptors [88,89]. These treatments all have a common complication: they form a complex with their targets. Cells may retain these complexes and form aggregates. Furthermore, once complexes form, the enzyme may no longer function as intended. Inhibition of certain enzymes results in accumulation of substrate precursors. The buildup of complexes and precursors can be detrimental to the physiology of the cells; thus, these complexes lead to side effects. Systematic disorders such as metabolic syndrome could be more effectively treated with a systematic (nuclear) regulator. 25HC3S epigenetically regulates the transcription of proteins and enzymes involved in lipid synthesis, inflammation, and apoptosis. Currently, the current data from DNA microarrays and computational analysis confirm that treatment with 25HC3S in vitro and or in vivo significantly regulates the expressions of numerous genes, which are mainly involved in inflammation, lipid metabolism, and apoptosis. 

Progression of the acute organ injury (AOI) can result in acute organ failure (AOF) including acute liver, lung, and kidney failure, leading to the sudden death of previously healthy individual. Globally, viral, drug-induced, and endotoxin-induced liver injury account for most cases of AOF. The pathophysiology of endotoxic shock consists of four development stages: 1. bacterial infection (sepsis), 2. endotoxin release such as LPS; 3. inflammatory response; and 4. shock, which can be followed by acute multiple organ failure. In 2013, sepsis was the second-most common reason for hospitalization, accounting for 3.6% of stays. Furthermore, sepsis accounted for more than $23.7 billion or 6.2% of total US hospital costs, making it number one in hospital costs. Early administration of 25HC3S/25HCDS has already been shown to significantly improve organ function and to decrease mortality in LPS-induced animal models [66], and they may have the potential be an effective therapeutic option for early stage sepsis. 

25HC3S has been tested in vitro (cells) and in vivo (animal models and preliminary data from NASH patients) [32,60,61,62,64,66,79,83,84]. Although additional testing needs to and will be done, current data all elucidate to the fact that 25HC3S/25HCDS have the potential to promote recovery of metabolic syndromes and may be able to prevent and/or treat acute organ failure. 

## Figures and Tables

**Figure 1 metabolites-11-00009-f001:**
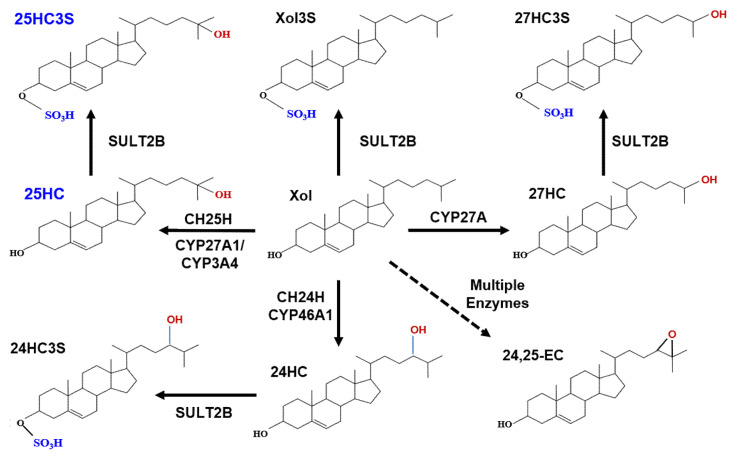
Metabolic pathway of oxysterol sulfates.

**Figure 2 metabolites-11-00009-f002:**
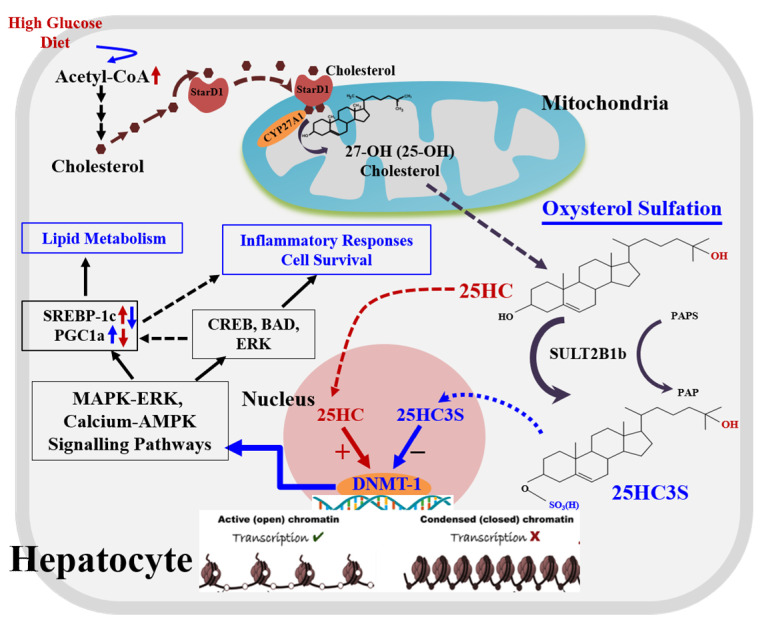
A novel regulatory pathway of oxysterol sulfation.

**Figure 3 metabolites-11-00009-f003:**
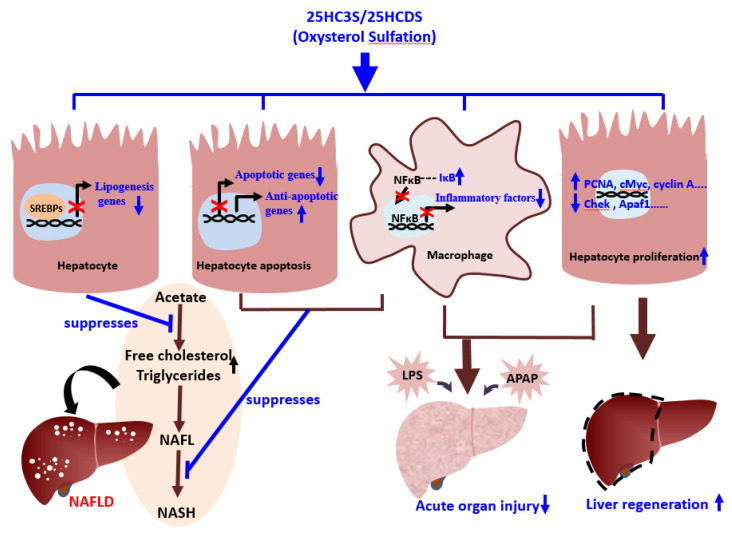
Schematic diagram of the role of oxysterol sulfation in nonalcoholic fatty liver diseases (NAFLD), acute organ injury, and liver regeneration.

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
