# Peer review of "Cholesterol Metabolites 25-Hydroxycholesterol and 25-Hydroxycholesterol 3-Sulfate Are Potent Paired Regulators: From Discovery to Clinical Usage"

_metabolites, 2020, doi:10.3390/metabo11010009_

Round 1

Reviewer 1 Report

In the review, authors explicitly summarized the current knowledge available about the roles of cholesterol metabolites with an extensive survey of references. Reviewer think that the review is well-written and will be useful for the general readers. To ensure the accuracy and completion of a paper, I would like to point out some minor point. 

  1. L151-152 “The function of 25HC and 25HC3S in global regulation indicates that they are epigenetic regulators.” and P6. L199-201 “25HC3S as an endogenous and orally bioavailable epigenetic regulator has been shown in nonclinical studies to play an important regulatory role in lipid homeostasis, inflammation, and cell survival, which have strong clinical significance as shown Fig. 3.”. As described in the manuscript, 25HC is typical LXR ligand and the influence of 25HC is basically considered as the result of activated LXR-dependent transcriptions or protein-protein interactions. Therefore, the information is of interest whether the 25HC-dependent functions described in the manuscript (such as DNA methylation, suppression of inflammation, apoptosis, hepatocyte regeneration or lipid regulations) were caused depending on LXR activations. Please consider to describe the information about the involvement of LXR in each function. The same is true of 25HC-sulfate.

  1. L222-228, “Higher oxysterol sulfate concentration resulted in... rather than clearance of lipid stores 18.”: It seems that authors were supposed to cite reference18 for these four sentences in L222-228 with adding [18] after the last sentences. Reviewer think that [18] should be after the first sentence or after the each sentence in L222-228. The same is true of L236-243 and L282-294.

  1. Authors may consider to add the reference in the sentences as listed below
    • L221-222, “Overexpression of SULT2B1b, in mice fed with high fat diet (HFD), significantly increased oxysterol sulfate formation in the liver.
    • L229-230 “Administration of 25HC3S reduced lipid accumulation, improved insulin resistance, and suppressed inflammation in vitro and in vivo.
    • L230-232 “Several studies have demonstrated pro-inflammatory cytokines, such as TNFa, IL-1 and IL-6, are the major contributors to the progression from steatosis to NASH.”

Author Response

Reviewer 1
Comments and Suggestions for Authors

In the review, authors explicitly summarized the current knowledge available about the roles of cholesterol metabolites with an extensive survey of references. Reviewer think that the review is well-written and will be useful for the general readers. To ensure the accuracy and completion of a paper, I would like to point out some minor point.

  1. L151-152 “The function of 25HC and 25HC3S in global regulation indicates that they are epigenetic regulators.” and P6. L199-201 “25HC3S as an endogenous and orally bioavailable epigenetic regulator has been shown in nonclinical studies to play an important regulatory role in lipid homeostasis, inflammation, and cell survival, which have strong clinical significance as shown Fig. 3.”. As described in the manuscript, 25HC is typical LXR ligand and the influence of 25HC is basically considered as the result of activated LXR-dependent transcriptions or protein-protein interactions. Therefore, the information is of interest whether the 25HC-dependent functions described in the manuscript (such as DNA methylation, suppression of inflammation, apoptosis, hepatocyte regeneration or lipid regulations) were caused depending on LXR activations. Please consider to describe the information about the involvement of LXR in each function. The same is true of 25HC-sulfate.

The reviewer raises a good question. It is true that 25HC has been believed as LXR ligand, activator of LXR, for several decades. However, LXR targeting molecules are very limited and can not be used to explain the function of 25HC global regulation in lipid metabolism, inflammatory responses, and cell apoptosis. Our recent publication indicates that 25HC specifically activates DNMT-1 activity by 8-fold at 3 uM levels and inhibits or activates many signaling pathways including LXR signaling pathway. The results give us a strong evidence that 25HC plays much more important roles than LXR signaling.

  1. L222-228, “Higher oxysterol sulfate concentration resulted in... rather than clearance of lipid stores 18.”: It seems that authors were supposed to cite reference18 for these four sentences in L222-228 with adding [18] after the last sentences. Reviewer think that [18] should be after the first sentence or after the each sentence in L222-228. The same is true of L236-243 and L282-294.

We have corrected the citation as suggested.

  1. Authors may consider to add the reference in the sentences as listed below
  • L221-222, “Overexpression of SULT2B1b, in mice fed with high fat diet (HFD), significantly increased oxysterol sulfate formation in the liver.”

We have added the reference as suggested.

  • L229-230 “Administration of 25HC3S reduced lipid accumulation, improved insulin resistance, and suppressed inflammation in vitro and in vivo.”

We have added the reference as suggested.

  • L230-232 “Several studies have demonstrated pro-inflammatory cytokines, such as TNFa, IL-1 and IL-6, are the major contributors to the progression from steatosis to NASH.”

We have added the reference as suggested.

Reviewer 2 Report

Wang and colleagues present a generally well-written, scholarly review of oxysterols and oxysterol sulfates and their roles in biology and pathophysiology.  In general, this review does a good job in synthesizing a complex literature for the readership.  I have the following comments:

  1. Some of the nomenclature used by the authors appears to be nonstandard, at least to this reviewer. For example, C25HL for cholesterol-25-hydroxylase (instead of CH25H) and C24HL for cholesterol 24-hydroxylase.  Unless the authors can clarify the source of this protein nomenclature, they are advised to stick to more standard terms/protein names.
  2. 1 is a bit misleading with regard to synthesis of 24,25-EC. This lipid is thought not to be an oxidation product of cholesterol (as implied by the figure), but to be a shunt side-product from an intermediate step in the cholesterol synthesis pathway. The authors should make this clearer in the figure and/or its legend.
  3. Cholesterol-25-hydroxylase is described by the authors on p. 2 as residing in cytosol, but most evidence suggests that it is ER-associated.   
  4. In section 2, the authors have left out some of the most prominent recent papers on 25HC regulation of inflammation. The following should be included (PMIDs: 24994901, 29033131, 25104388).
  5. While the writing is generally very good, several syntactical/spelling errors were noted. A careful editing by an English language expert is recommended.

Author Response

Reviewer 2

Wang and colleagues present a generally well-written, scholarly review of oxysterols and oxysterol sulfates and their roles in biology and pathophysiology. In general, this review does a good job in synthesizing a complex literature for the readership. I have the following comments:

  1. Some of the nomenclature used by the authors appears to be nonstandard, at least to this reviewer. For example, C25HL for cholesterol-25-hydroxylase (instead of CH25H) and C24HL for cholesterol 24-hydroxylase. Unless the authors can clarify the source of this protein nomenclature, they are advised to stick to more standard terms/protein names.
  2. We have corrected the nomenclature as reviewer’s suggestion in the figures and in the text.
  1. 1 is a bit misleading with regard to synthesis of 24,25-EC. This lipid is thought not to be an oxidation product of cholesterol (as implied by the figure), but to be a shunt side-product from an intermediate step in the cholesterol synthesis pathway. The authors should make this clearer in the figure and/or its legend.

We have added a reference for the synthesis of 24,25-EC after the sentence “Cholesterol precursor can be used for synthesis of desmosterol via a shunt of the mevalonate pathway. The desmosterol can be oxygenated by CYP46A1 to form 24, 25-epoxycholesterol (24,25EC)”.

  1. Cholesterol-25-hydroxylase is described by the authors on p. 2 as residing in cytosol, but most evidence suggests that it is ER-associated.  

The reviewer is correct. We have corrected the statement that CH25H is ER-associated.

  1. In section 2, the authors have left out some of the most prominent recent papers on 25HC regulation of inflammation. The following should be included (PMIDs: 24994901, 29033131, 25104388).

Good suggestion. We have added all these references in the statement that 25HC plays important role in the inflammatory responses.

  1. While the writing is generally very good, several syntactical/spelling errors were noted. A careful editing by an English language expert is recommended.

We have tried our best to carefully edit the manuscript. It looks much better now.

Reviewer 3 Report

This review is devoted to cholesterol metabolites following hydroxylation and sulfation reactions and their role in in vitro and in vivo experiments of disease development and recovery. Just few suggestions/corrections before approval:

Title: 25HC is 25-hydroxycholesterol, so I cannot understand 25HC-hydroxycholesterol

line 23: correct "25-hyrdoxy"

line 32: change to "steroid hormones"

Figure 1: please change to "O-SO3H" in the structures

lines 63-64: maybe this metabolite must be added into the figure 1

lines 88-90. revise the sentence

lines 128 and 198 versus lines 274: is the subject singular or plural?

line 140: correct "singling"

line 164: change to "epigenetically regulate"

line 168: change to "is responsible"

text in general: please make uniform the American or English spelling, e.g. signalling or signaling

line 270: correct "This signify"

lines 324, 326, and 328: change to "of the greatest effect"

line 346: change to "epigenetically regulates"

line 359: change to "has already shown"

lines 415 and 416: "Xol3S: Cholesterol 3-sulfate" is repeated twice

Author Response

Reviewer 3

This review is devoted to cholesterol metabolites following hydroxylation and sulfation reactions and their role in in vitro and in vivo experiments of disease development and recovery. Just few suggestions/corrections before approval:

Title: 25HC is 25-hydroxycholesterol, so I cannot understand 25HC-hydroxycholesterol

We have corrected as the 25-hydroxycholesterol

line 23: correct "25-hyrdoxy"

We have corrected

line 32: change to "steroid hormones"

We have changed to “steroid hormones”

Figure 1: please change to "O-SO3H" in the structures

We have changed all O-SO3(H) to be O-SO3H in the Figure 1

lines 63-64: maybe this metabolite must be added into the figure 1

34,25-epoxycholeleterol sulfate is missing in the Figure 1 because we do not believe this may happen in vivo.

lines 88-90. revise the sentence

25HC plays an important role in lipid accumulation in hepatocytes and development of non-alcoholic fatty liver diseases (NAFLD). 25HC regulates excess acetyl CoA for synthesis of lipids as energy storage. For example, carbohydrates such as fructose and glucose, can be hydroxylated to acetyl-CoA, the major substrate for producing energy, ATP, in vivo. When ATP concentrations reach high levels, decreases in ratio of AMP/ATP, the excess acetyl-CoA is shunted into synthesis of cholesterol and oxysterol such as 25HC. 25HC induces biosynthesis of free fatty acids and triglycerides.

lines 128 and 198 versus lines 274: is the subject singular or plural?

The reviewer is correct. We corrected all these mistakes.

line 140: correct "singling"

We have made it as signalling

line 164: change to "epigenetically regulate"

We have changed to “epigenetically regulate”

line 168: change to "is responsible"

We have changed to “is responsible”

line 270: correct "This signify"

We rewritten the sentence as “, indicating that SULT2B1b is responsible for the expression”

lines 324, 326, and 328: change to "of the greatest effect"

We have changed them to “of the greatest effect”

line 346: change to "epigenetically regulates"

We have changed to “epigenetically regulates”

line 359: change to "has already shown"

We have changed to “has already shown”

lines 415 and 416: "Xol3S: Cholesterol 3-sulfate" is repeated twice

We have deleted one of them.

Thanks to this reviewer. Helps a lot. We have corrected all these errors as “track changes”.